# Using Chatbots as AI Conversational Partners in Language Learning

**Jose Belda-Medina *** and **José Ramón Calvo-Ferrer**

Digital Language Learning (DL2) Research Group, University of Alicante, 03690 Alicante, Spain
* Correspondence: jr.belda@ua.es; Tel.: +34-965-909-438

**Abstract:** Recent advances in Artificial Intelligence (AI) and machine learning have paved the way for the increasing adoption of chatbots in language learning. Research published to date has mostly focused on chatbot accuracy and chatbot–human communication from students' or in-service teachers' perspectives. This study aims to examine the knowledge, level of satisfaction and perceptions concerning the integration of conversational AI in language learning among future educators. In this mixed method research based on convenience sampling, 176 undergraduates from two educational settings, Spain ($n = 115$) and Poland ($n = 61$), interacted autonomously with three conversational agents (Replika, Kuki, Wysa) over a four-week period. A learning module about Artificial Intelligence and language learning was specifically designed for this research, including an ad hoc model named the Chatbot–Human Interaction Satisfaction Model (CHISM), which was used by teacher candidates to evaluate different linguistic and technological features of the three conversational agents. Quantitative and qualitative data were gathered through a pre-post-survey based on the CHISM and the TAM2 (technology acceptance) models and a template analysis (TA), and analyzed through IBM SPSS 22 and QDA Miner software. The analysis yielded positive results regarding perceptions concerning the integration of conversational agents in language learning, particularly in relation to perceived ease of use (PeU) and attitudes (AT), but the scores for behavioral intention (BI) were more moderate. The findings also unveiled some gender-related differences regarding participants' satisfaction with chatbot design and topics of interaction.

**Keywords:** chatbots; intelligent conversational agents; language learning; pre-service teachers; perceptions and satisfaction

## 1. Introduction

Nowadays, there is a growing interest in Artificial Intelligence (AI) and chatbots, which have been widely adopted in different areas such as e-commerce, healthcare and education [1,2]. Chatbot technology has rapidly evolved over the last decades, partly thanks to modern advances in Natural Language Processing (NLP) and machine learning [3,4]. The history of chatbots can be traced back to the 1950s when Alan Touring formulated his complex question 'Can machines think?', published as an article in *Computing Machinery and Intelligence* (1950). Since then, a good amount of chatbots have emerged such as Eliza (1966), Parry (1972), Racter (1983), Jabberwacky (1988) and A.L.I.C.E. (1995), some of them still in use today. These chatbots were originally text-based and preset, based on Q&A scripts, so their responses were considered predictable, and the interaction was not perceived as natural by human standards.

However, modern chatbots have incorporated new functionalities such as R&S technologies (voice recognition and synthesis), customized interaction, integration with third-party apps, omnichannel deployment, context-awareness and multi-turn capability [3,5,6]. As a result, there is today a wide range of chatbots integrated in all shapes and forms into different electronic devices, programs and applications, for example messaging apps (Whatsapp, Telegram, Kik, Slack), video games and gaming platforms (Xbox, Roblox) and

social networks (Instagram, Facebook Messenger, Twitter). In fact, there are virtually very few areas where chatbots are still not or will not be present in some form in the near future. As Fryer et al. [7] pointed out:

> 'During their first three decades, chatbots grew from exploratory software, to the broad cast of potential friends, guides and merchants that now populate the internet. Across the two decades that followed this initial growth, advances in text-to-speech-to-text and the growing use of smartphones and home assistants have made chatbots a part of many users' day-to-day lives'. (p. 17)

The term chatbot can be misleading as it may refer to a wide variety of programs used in different formats and with different purposes. Generically, a chatbot could be defined as a computer program based on AI that simulates human conversation via a textual and/or auditory method. However, there are different chatbot-related concepts used nowadays which do not have the same meaning: chatter bot, smart bot, educabot, quizbot, digital assistant, personal assistant, virtual tutor, conversational agent, etc. [8]. Broadly speaking, chatbots could be classified depending on the criteria shown in Table 1.

**Table 1.** Chatbot classification.

| Interaction Mode | Deployment | Knowledge | Service Provided | Purpose | Complexity | Input Processing |
|---|---|---|---|---|---|---|
| text-based (and/or) voice-enabled | web-based app-integrated standalone tools | open domain closed domain | interpersonal intrapersonal inter-agent | informative task-based conversational | menu/button based keyword recognition contextual | rule-based generative model |

Conversational AI, also known as conversational agents, can be described as smart software programs that learn through communication like humans; they originally gave standardized answers to typical questions but later evolved into more sophisticated programs thanks to Natural Language Understanding (NLU), neural networks and deep-learning technologies. They constitute an advanced dialog system which simulates conversation in writing and/or speech with human users, typically over the Internet. They can be text-based and/or read from and respond with speech, graphics, virtual gestures or haptic-assisted physical gestures. Additionally, some of them have been trained to perform certain tasks, also known as IPAs (Intelligent Personal Assistants), for example IBM Watson, Apple Siri, Samsung Bixby and Amazon Alexa [9].

Conversational AI could become useful tools as language-learning partners as well as tireless tutors available anytime and anywhere, particularly in contexts and settings where formal education and access to native speakers are not an option. Works published to date have focused mainly on the accuracy of this form of chatbot and the benefits and attitudes among current students and in-service teachers [10,11] but research about the perceptions concerning these programs among future educators is scant. This is essential as pre-service teachers will be responsible for not only adopting but also integrating these conversational agents in the classroom. To bridge this gap, this research aims to examine the knowledge, level of satisfaction and perceptions concerning the use of conversational partners in foreign language learning among teacher candidates.

## 2. Chatbots in Language Learning

There is an increasing amount of works about the integration of conversational agents in language learning, which can be framed within the linguistic area known as ICALL (Intelligent Computer-Assisted Language Learning). Fryer and Carpenter [12] analyzed the interaction between two different chatbots with 211 students and concluded that they could be effectively used for self-practice via casual conversation, although the authors believed these programs are more useful for advanced language students.

Similarly, Hill et al. [13] compared 100 instant messaging conversations to 100 exchanges with the chatbot named Cleverbot along seven different dimensions: words per message, words per conversation, messages per conversation, word uniqueness and use of profanity, shorthand and emoticons. The authors found some notable differences such as that people communicated with the chatbot for longer durations (but with shorter messages) than they did with another human, and that human–chatbot communication lacked much of the richness of vocabulary found in conversations among people, which is consistent with later works related with the use of chatbots among children [14]. Ayedoun et al. [15] used a semantic approach to demonstrate the positive impact of using a conversational agent on the willingness to communicate (WTC) in the context of English as a Foreign Language (EFL) by providing users with various daily conversation contexts. In their study, the conversational agent facilitated an immersive environment that enabled learners to simulate various daily conversations in English in order to reduce their anxiety and increase their self-confidence.

Several reviews about conversational agents in language learning have been published to date. Io and Lee [16] identified some research gaps associated with 'the dearth of research from the human point of view' (p. 216), and concluded that modern forms of chatbot such as mobile chat apps, embedded website services or wearable devices should be given special attention in future research. Shawar [17] revised the adoption of different chatbots in language learning and highlighted some of its benefits such as student enjoyment, decreased language anxiety, opportunities of endless repetitions and multimodal capabilities.

In another review, Radziwill and Benton [18] examined some attributes and quality issues related with chatbot development and implementation. The authors took into consideration factors such as efficiency, effectiveness and satisfaction, and concluded that chatbots can positively support learning but also be used to engineer social harm (misinformation, rumors). Similarly, Haristani [19] analyzed different types of chatbots, and pointed out six advantages: decreased language anxiety, wide availability, multimodal practice, novelty effect, rich variety of contextual vocabulary and effective feedback. Bibauw et al. [20,21] presented a systematic review of 343 publications related with dialogue-based systems. The authors described the interactional, instructional and technological aspects, and underlined the learning gains in vocabulary and grammatical outcomes as well as the positive effects on student self-confidence and motivation.

More recently, Huang et al. [22] inspected 25 empirical studies and identified five pedagogical uses of chatbots: as interlocutors, as simulations, as helplines, for transmission and for recommendation. The authors underlined certain affordances such as timeliness, ease of use and personalization, as well as some challenges such as the perceived unnaturalness of the computer-generated voice, and failed communication. Similarly, Dokukina and Gumanova [23] highlighted the novelty of using chatbots in a new scenario based on microlearning, automation of the learning process and an adaptive learning environment.

However, several drawbacks about the educational use of chatbots have also been reported. Criticism is mainly based on the scripted nature of their responses, which make them rather predictable, their limited understanding (vocabulary range, intentional meaning) and the ineffective communication (off topics, meaningless sentences) [12]. The two most cited challenges are related with students' (lack of) interest in such predictable tools, and the (in)effectiveness of chatbot–human communication in language learning. In the first case, Fryer et al. [24] compared chatbot–human versus human–human conversations through different tasks and evaluated the impact on the student interest. The authors concluded that human partner tasks can predict future course interest among students but this interest declines under chatbot partner conditions. In other words, student interest in chatbot conversation decreases after a certain period of time, usually known as the novelty effect.

Concerning the linguistic (in)effectiveness of chatbot–human communication, Coniam [25,26] examined the accuracy of five renowned chatbots (Dave, Elbot, Eugene,

George and Julie) at the grammatical level from an English as a Second Language (ESL) perspective. The author noticed that chatbots often provide meaningless answers, and observed some problems related with the accuracy rate for the joint categories of grammar and meaning. According to Coniam, chatbots do not yet make good chatting partners but improvements in their performance can facilitate future developments.

Interest in chatbots has grown rapidly over the last decade in light of the increasing number of publications [27], which lead some authors to talk about a chatbot hype [28,29]. However, the benefits seem to outweigh the limitations since AI is a fast-growing technological sector, and many authors strongly believe in its educational potential. For example, Kim [30] recently investigated the effectiveness of different chatbots in relation to four language skills (listening, reading, speaking and writing) and vocabulary, and pointed out that chatbots can become valuable tools to provide EFL students with opportunities to be exposed to authentic and natural input through both textual and auditory methods.

Research on this topic has mainly focused on the accuracy of chatbot–human communication [7,23,31], language learning gains [21,32] and satisfaction [33–35] from the current students' perspective; and on the attitudes toward the integration of chatbots in education from the in-service teachers' perspective [36,37]. In this sense, Chen et al. [38] investigated the impact of using a newly developed chatbot to learn Chinese vocabulary and students' perception through the technology acceptance model (TAM), and concluded that perceived usefulness (PU) was a predictor of behavioral intention (BI) whereas perceived ease of use (PEU) was not.

Regarding in-service teachers, Chocarro et al. [37] examined the acceptance of chatbots among 225 primary and secondary education teachers using also the technology acceptance model (TAM). The authors correlated the conversational design (use of social language and proactiveness) of chatbots with the teachers' age and digital skills, and their results showed that perceived usefulness (PU) and perceived ease of use (PEU) lead to greater chatbot acceptance. Similarly, Chuah and Kabilan [36] analyzed the perception of using chatbots for teaching and learning delivery in a mobile environment among 142 ESL (English as a second language) teachers. According to the authors, the in-service teachers believed chatbots can be used to provide a greater level of social presence, which eventually creates an environment for the students to be active in a second language.

However, research about pre-service teachers' perceptions concerning chatbot integration in language learning is scant. A notable exception is the study of Sung [39], who evaluated 17 AI English-language chatbots developed by nine groups of pre-service primary school teachers using the Dialogflow API. The results showed that pre-service teachers found chatbots useful to engage students in playful and interactive language practice. More recently, Yang [40] examined the perceptions of 28 pre-service teachers on the potential benefits of employing AI chatbots in English instruction and its pedagogical aspects. The authors concluded that these programs can enhance learners' confidence and motivation in speaking English and be used to supplement teachers' lack of English competency in some contexts.

The integration of chatbots as language-learning partners will depend to a certain extent on the pre-service teachers' knowledge and willingness to adopt them as future educators, so there is a need to examine their awareness of this technology. The novelty of this study is that it aims to evaluate the knowledge and perceptions concerning the use of conversational agents among language teacher candidates as well as their level of satisfaction through an ad hoc model, the Chatbot–Human Interaction Satisfaction Model (CHISM).

## 3. Research Questions

The four research questions are:

- What knowledge do language teacher candidates have about chatbots?

- What is their level of satisfaction with the three conversational agents selected (Replika, Kuki, Wysa) regarding certain linguistic (semantic coherence, lexical richness, error correction, etc.) and technological features (interface, design, etc.)?
- What are their perceptions toward the integration of conversational agents in language learning as future educators?
- What are the effects of the educational setting and gender on the results of the previous questions?

## 4. Context, Method and Materials

### 4.1. Context and Sampling

This research was based on convenience sampling; all participants ($n$ = 176) were teacher candidates enrolled in two similar subjects on applied linguistics taught concurrently in two different settings: the University of Alicante (Spain $n$ = 115) and the Silesian University of Technology (Poland $n$ = 61). The original sample size was 185 but nine students dropped the class. They were all junior college students in Education with a B2–C1 (CEFR) level of English; a few of them were English native speakers. Gender distribution was similar in both settings, Spain (m = 19%, f = 81%) and Poland (m = 20%, f = 80%), and 98% of the participants were aged between 20 and 29. The two subjects on applied linguistics were taught by the researchers during the Spring term of 2022; the classes consisted of two-hour sessions that ran twice a week along four consecutive months; one month was dedicated to language learning and Artificial Intelligence (AI). This study is part of a larger research project, the application of AI and chatbots to language learning, financed by the Instituto de Ciencias de la Educación at the Univesity of Alicante (Reference number: 5498). The overall project adheres to ethical principles set out by both institutions under specific regulations (see last section about consent) regarding the requirements related to information, consent, anonymity and the right to withdraw from the project. All participants gave written consent to use the data obtained for scientific purposes and their names were omitted to ensure anonymity.

### 4.2. Materials and Procedure

In this one-month project, a learning module entitled 'The integration of AI and chatbots in language learning' was specifically designed on Moodle for both groups. This module included a short introduction to the topic and several materials (three videos and an article about chatbots in language learning) that all participants had to watch and read during the first week of the project, and they also had to complete a questionnaire based on the materials. The module contained a template analysis (TA) including links to the three chatbots and detailed instructions about the human–chatbot interaction, which is described in the following section (instruments). The participants were instructed to interact with the three conversational agents as future educators and not as English language learners. Therefore, they should carefully follow the instructions provided in the TA (use different registers, sentence structures, intentional mistakes, irony, etc.). This out-of-class interaction should take place over the four-week period on a regular basis. To ensure this regularity, the participants needed to take notes during their interaction together with 10 screenshots with the corresponding date as part of the template analysis (TA), which they needed to submit during the final week. The research comprised four stages as shown in Figure 1.

Three conversational AI were employed. The first one named Kuki, originally Mitsuku and available on www.kuki.ai, was created by Pandorabots in 2005 in the UK. This chatbot has been awarded several Loebner prizes (2013, 2016, 2017, 2018, 2019) and uses the Artificial Linguistic Internet Computer Entity database. It is linked to Twitch, Telegram, Facebook, Kik and Discord and it can also be found on Youtube and the Roblox game. Kuki uses an avatar featuring an 18-year-old female from Leeds (UK). It offers some special features such as the option to arrange video calls.

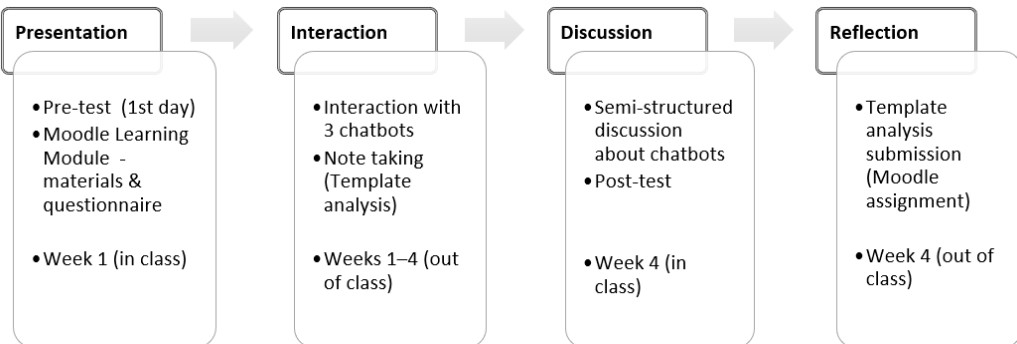

**Figure 1.** Research stages.

The second chatbot named Replika and available on replika.com was created by the startup Luka in San Francisco (USA) in 2017. This bot has an innovative design as users need to customize their own avatar when they first create their Replika account. Therefore, the participants could personalize their avatars among a range of options (gender, hairstyle, outfit, etc.). Replika gets feedback from conversation with the users as well as access to social networks if permission is granted, so the more the users talk to the bot the more it learns from them, and the more human it looks. The bot has memory, keeps a diary and the users can gain more experience to reach different levels (scalable). As a special feature, it includes an augmented-reality-based option, so participants can launch the AR-based avatar to chat with the Replika.

The third chatbot called Wysa and available on www.wysa.io was created by the startup Touchkin eServices in Bangalore (India) in 2016. This is a specialized type of program as it is an AI-based emotionally intelligent bot that uses evidence-based cognitive-behavioral techniques (CBTs) for wellbeing purposes. It works as a psychotherapist and the avatar is a pocket penguin that functions as a well-being coach. This chatbot can recognize over 70 different emotion subtypes (boredom, anxiety, sadness, anger, depression, etc.). Wysa is used by some healthcare companies and organizations around the world, such as the National Health Service (UK), Aetna and Accenture. Figure 2 illustrates three examples of chatbot interaction as provided by the teacher candidates.

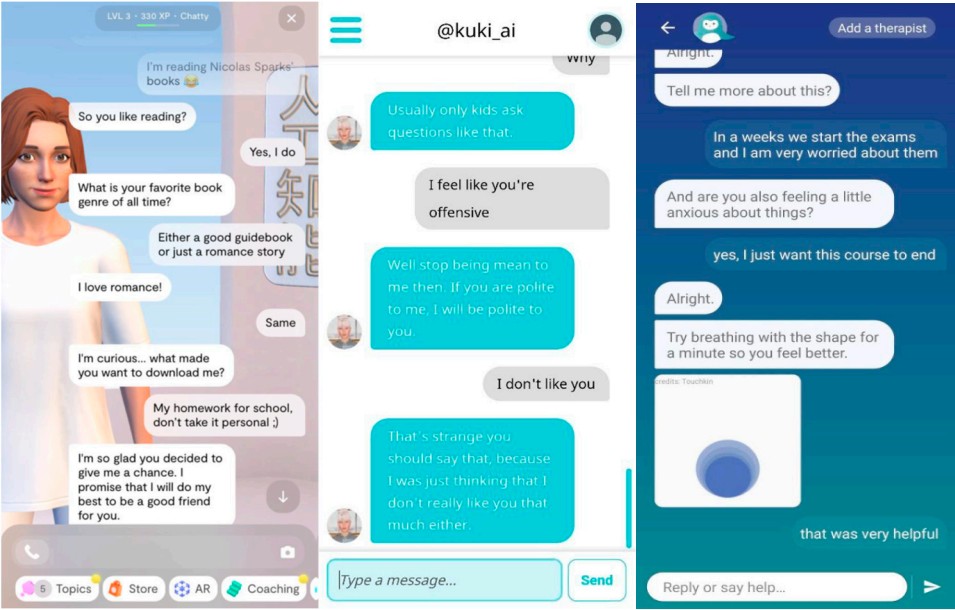

**Figure 2.** Screenshots of the interaction with the three conversational agents (from left to right: Replika, Kuki, Wysa).

*4.3. Method and Instruments*

This mixed method research followed a sequential explanatory design (two-phase model) as described by Pardede for EFL studies [41]. Quantitative data were collected through two surveys before (pre) and after (post) the treatment. The pre-survey (ANNEX A in Supplementary Materials) included eight questions arranged in two sections: socio-demographic data and prior knowledge on chatbots. The post-survey (ANNEX B in Supplementary Materials) was organized in two sections: the first one included an ad hoc model, the Chatbot–Human Interaction Satisfaction Model (CHISM), aimed at measuring participants' levels of satisfaction with the three conversational agents, and it contained 15 items associated with different linguistic, technological and experiential features. The second section was based on the technology acceptance model (TAM), which was first introduced by Fred Davis in 1986 [42]; we used an adapted version of the TAM2 model [43]. This model integrates 25 items organized in seven dimensions: perceived ease of use (PEU), perceived usefulness (PU), attitude (ATT), perceived behavior control (PBC), behavioral intention (BI), self-efficacy (SE) and personal innovativeness (PI).

Qualitative data were gathered through written reports based on a template analysis (TA) as defined by King [44], which was provided to the teacher candidates during the first day (ANNEX C in Supplementary Materials). The TA included two sections: the first one contained detailed instructions about the human–chatbot interaction; the second comprised open-ended questions about the benefits and limitations of each conversational partner. The participants had to respond to those questions and submit answers to the instructors including 10 screenshots with the corresponding date to ensure the frequency of interaction. Quantitative and qualitative data were analyzed through the IBM SPSS Statisics 22 and the QDA Miner software, respectively.

## 5. Results and Discussion

The pre-survey yielded similar results in terms of technology ownership (computer, laptop, tablet, smartphone, chatbot) for both groups, Spanish (M = 3.07, SD = 0.917) and Polish (M = 2.82, SD = 0.719). However, there were some differences about prior knowledge of chatbots as shown in Table 2. Spanish participants seemed to be more familiar with Intelligent Personal Assistants (IPAs), except for Samsung Bixby.

**Table 2.** Familiarity with modern chatbots (Intelligent Personal Assistants).

|  | Alexa (Amazon) | | Cortana (Microsoft) | | Siri (Apple) | | Google Assistant | | Watson (IBM) | | Bixby (Samsung) | |
|---|---|---|---|---|---|---|---|---|---|---|---|---|
|  | Sp. | Pol. | Sp. | Pol. | Sp. | Pol. | Sp. | Pol. | Sp. | Pol. | Sp. | Pol. |
| Knowledge | 96% | 70% | 72% | 25% | 98% | 86% | 87% | 76% | 2% | 0% | 19% | 28% |

The pre-survey results also evidenced that both groups rated similarly the overall usefulness of chatbots in different areas (Sp. M = 2.53 and Pol. M = 2.54) as illustrated in Table 3.

**Table 3.** Perceived usefulness of chatbots (five-point scale: from 1 = not useful at all to 5 = very useful).

|  | E-Commerce | | Education | | Healthcare | | Psychological | | Social Networking | | General Information | |
|---|---|---|---|---|---|---|---|---|---|---|---|---|
|  | Sp. | Pol. | Sp. | Pol. | Sp. | Pol. | Sp. | Pol. | Sp. | Pol. | Sp. | Pol. |
| usefulness | 2.1 | 2.5 | 2.6 | 2.7 | 2.1 | 2.0 | 2.1 | 1.9 | 2.7 | 2.5 | 3.4 | 3.4 |

Concerning frequency of interaction with the three conversational partners, the post-survey elicited similar results for both groups, with an average of one hour a day during the four-week period (Spanish M = 1.2 h/day vs. Polish M = 1.1 h./day). Interestingly, both groups interacted more often with Replika, followed by Kuki and finally Wysa as shown in

Table 4, though no specification had been provided. This frequency may actually reflect the participants' interest in each chatbot depending on certain factors explained later in the qualitative results.

**Table 4.** Frequency of interaction (Choices: <1 h, 1–3 h, 3–5 h, 5+ h per week).

|  | Replika | | Kuki | | Wysa | |
|---|---|---|---|---|---|---|
|  | **M** | **SD** | **M** | **SD** | **M** | **SD** |
| Spanish | 2.9 h. | 0.871 | 3.3 h. | 0.993 | 2.7 h. | 0.840 |
| Polish | 2.6 h. | 0.564 | 3.2 h. | 0.783 | 2.3 h. | 0.467 |

A model was specifically designed to measure the participants' level of satisfaction with the three conversational partners, named the Chatbot–Human Interaction Satisfaction Model (CHISM) and shown in Table 5. This model comprised 15 items distributed in three dimensions: the first one was related to linguistic features (items 1–9), the second was associated with technological features (items 10–12) and the last one was about user engagement, enjoyment and further interest (items 13–15). This model was based on a five-point Likert scale with a reliability coefficient of 0.957 (Cronbach's alpha). During the first week of instruction, the participants were told how to interpret each of those items. Table 5 displays the CHISM results for both groups. The highest scores were obtained by Replika, particularly in design (M Sp. = 4.1 and Pol. = 4.5) and interface (M Sp. = 4.0 and Pol. = 4.5).

**Table 5.** Results of the Chatbot–Human Interaction Satisfaction Model (CHISM). Five-point scale (1 = not satisfied at all to 5 = completely satisfied).

| $n = 176$ $\alpha = 0.957$ | Replika | | | | Kuki | | | | Wysa | | | |
|---|---|---|---|---|---|---|---|---|---|---|---|---|
|  | Sp | | Pol. | | Sp | | Pol. | | Sp | | Pol. | |
|  | **M** | **SD** | **M** | **SD** | **M** | **SD** | **M** | **SD** | **M** | **SD** | **M** | **SD** |
| 1. Semantic coherent behavior | 3.6 | 1.11 | 4.1 | 0.69 | 3.1 | 1.15 | 3.0 | 1.12 | 3.3 | 1.22 | 3.0 | 0.97 |
| 2. Sentence length and complexity | 3.8 | 1.06 | 4.2 | 0.71 | 3.3 | 1.04 | 3.4 | 0.97 | 3.3 | 1.14 | 3.2 | 0.92 |
| 3. R&S technologies | 3.4 | 1.06 | 3.4 | 1.02 | 3.3 | 1.11 | 3.5 | 1.09 | 3.4 | 1.19 | 3.1 | 0.96 |
| 4. Lexical richness | 3.9 | 1.08 | 4.3 | 0.74 | 3.2 | 1.16 | 3.3 | 1.16 | 3.5 | 1.22 | 3.2 | 1.12 |
| 5. Grammatical accuracy | 3.6 | 1.22 | 4.0 | 0.98 | 2.4 | 1.10 | 2.5 | 1.19 | 2.4 | 1.05 | 2.4 | 0.88 |
| 6. Error detection and correction | 2.5 | 1.15 | 2.7 | 1.14 | 3.1 | 1.07 | 2.8 | 0.88 | 3.1 | 1.11 | 2.9 | 0.75 |
| 7. Natural conversational interaction | 3.9 | 1.19 | 4.3 | 0.79 | 3.2 | 1.22 | 3.2 | 1.05 | 3.1 | 1.23 | 2.9 | 0.93 |
| 8. Chatbot response interval | 3.9 | 1.07 | 4.3 | 0.84 | 3.6 | 1.22 | 3.7 | 1.01 | 3.5 | 1.16 | 3.3 | 1.17 |
| 9. Non-verbal language | 3.6 | 1.17 | 4.1 | 1.00 | 3.2 | 1.31 | 3.2 | 1.03 | 2.9 | 1.13 | 3.0 | 1.01 |
| 10. Multimedia content | 3.7 | 1.18 | 4.3 | 0.87 | 3.4 | 1.19 | 3.3 | 1.17 | 3.0 | 1.18 | 3.3 | 1.17 |
| 11. Design | 4.1 | 1.12 | 4.5 | 0.67 | 3.5 | 1.15 | 3.3 | 1.08 | 2.9 | 1.16 | 3.3 | 1.22 |
| 12. Interface | 4.0 | 1.10 | 4.5 | 0.64 | 3.5 | 1.11 | 3.7 | 0.98 | 3.5 | 1.20 | 3.7 | 1.07 |
| 13. Engagement | 3.9 | 1.16 | 4.3 | 0.81 | 3.1 | 1.24 | 3.1 | 1.15 | 3.2 | 1.28 | 3.1 | 1.10 |
| 14. Enjoyment | 3.7 | 1.20 | 4.2 | 0.99 | 3.0 | 1.27 | 2.7 | 1.35 | 3.1 | 1.27 | 3.1 | 1.13 |
| 15. Further interest | 3.5 | 1.29 | 3.9 | 1.13 | 2.7 | 1.24 | 2.3 | 1.32 | 2.9 | 1.41 | 2.7 | 1.18 |
| Total | 3.7 | 1.14 | 4.1 | 0.87 | 3.2 | 1.17 | 3.1 | 1.10 | 3.1 | 1.20 | 3.1 | 1.04 |

On the linguistic level, the participants highly appraised the lexical richness (#4 M Sp. = 3.9 and Pol. = 4.3) of Replika, such as the use of different registers and the chatbot familiarity with colloquial expressions. The number of prearranged responses and off topics was perceived to be limited, as shown in semantic coherent behavior (#1 M Sp. = 3.6 and Pol. = 4.1). Regarding sentence length, the results were also positive (#2 M Sp. = 3.8 and Pol. = 4.2); the participants indicated that Replika combined several clauses with different

levels of complexity in the conversation. The response interval was considered to be quite natural in both groups (#8 M Sp. = 3.9 and Pol. = 4.3).

The only drawback mentioned was that Replika did not correct errors during the conversation (#6 M Sp. = 2.5 and Pol. = 2.7) as often as Kuki did (M Sp. = 3.1 and Pol. = 2.8), which could be a limitation for its use in language learning, particularly among lower-level students. This possible limitation was already indicated in previous works [45]. However, the teacher candidates were quite engaged (#13 M Sp. = 3.9 and Pol. = 4.3) and enjoyed the conversation with Replika (#14 M Sp. = 3.7 and Pol. = 4.2), which confirms previous results about EFL students' level of satisfaction with this conversational partner [46].

The other two chatbots, Kuki (M Sp. = 3.2 and Pol. = 3.1) and Wysa (M Sp. and Pol. = 3.1), reported similar overall scores, although Wysa was perceived to be more predictable due to its preset design. Nonetheless, the teacher candidates highly valued the innovativeness and usefulness of interacting with Wysa for their mental well-being, which is consistent with previous studies [47,48].

The effects of gender and educational setting on participants' levels of satisfaction were analyzed through the Mann–Whitney U test as the nonparametric alternative to the independent *t*-test for ordinal data. The only statistically significant difference observed was related to gender and chatbot design ($p = 0.009$), as shown in Table 6. Generally, female participants ($n = 142$) seemed to be more perceptive than male participants ($n = 34$) concerning the customizing options of some chatbots, as later explained in the qualitative results. In fact, this difference was evidenced in the case of Replika, which allowed avatar personalization (gender, race, name, etc.): 78% of the male teacher candidates opted for creating a female avatar; however, the preferences among female participants were more diverse: 59% of them created a female Replika, 37% chose a male Replika, and the remaining 4% opted for a non-binary avatar.

**Table 6.** Effect of gender and educational setting on the satisfaction results (CHISM).

| Participants' Gender | Mann–Whitney U | Z | Sig. (2-Tailed) | Educational Setting | Mann–Whitney U | Z | Sig. (2-Tailed) |
|---|---|---|---|---|---|---|---|
| Linguistic dimension (1–9) | 3,228,000 | −1.050 | 0.294 | Linguistic dimension (1–9) | 4,649,500 | −0.261 | 0.794 |
| Design dimension (10–12) | 2,659,000 | −2.624 | 0.009 | Design dimension (10–12) | 4,181,500 | −1.388 | 0.165 |

With regards to participants' perceptions concerning the use of chatbots in language learning, the TAM2 results were quite similar for both groups, Spanish and Polish, as illustrated in Table 7. The results for PEU (Sp. M = 3.6 and Pol. M = 3.7) and PU (Sp. M = 3.5 and Pol. M = 3.6) were positive, which confirms prior research about the perceived usefulness and easiness of chatbots among in-service teachers [37]. However, the scores for BI (Sp. and Pol. M = 2.9) and PBC (Sp. M = 3.0 and Pol. M = 2.9) were more moderate concerning pre-service teachers' future intention to use chatbots, which may partly contradict some previous works [39,40]. Although the participants showed interest in learning more about chatbots, they seemed to prioritize human to human over human to chatbot communication, which is in line with Fryer et al. [24].

The results of the other dimensions evidenced small differences with slightly higher scores among the Polish participants, as in SE (Sp. M = 3.4 and Pol. M = 3.6) and ATT (Sp. M = 3.5 and Pol. M = 3.7). In line with Chuah and Kabilan [36], the teacher candidates showed a moderate confidence and positive attitudes toward the integration of chatbots in language learning.

**Table 7.** Perceptions concerning the use of chatbots in language learning (TAM2 model). Dimensions: perceived ease of use (PEU), perceived usefulness (PU), usability (US), perceived behavior control (PBC), attitude (ATT), behavioral intention (BI), self-efficacy (SE) and personal innovativeness (PI).

| TAM2  $n = 176$  $\alpha = 0.914$ | Spanish ($n = 115$) | | Polish ($n = 61$) | |
|---|---|---|---|---|
| Items and Dimensions | M | SD | M | SD |
| 1. (PEU) I find chatbots easy to use | 3.9 | 0.928 | 4.3 | 0.593 |
| 2. (PEU) Learning how to use chatbots is easy for me | 3.9 | 0.858 | 4.3 | 0.606 |
| 3. (PEU) It is easy to become skillful at using chatbots in language learning | 3.5 | 0.926 | 3.4 | 0.788 |
| 4. (PEU) I find chatbots in language learning to be flexible to interact with | 3.4 | 0.941 | 3.4 | 0.901 |
| 5. (PEU) The interaction with chatbots in language learning is clear and understandable | 3.3 | 0.891 | 3.2 | 0.998 |
| 6. (PU) Using chatbots in language learning would increase the students' learning performance | 3.4 | 1.018 | 3.4 | 0.848 |
| 7. (PU) Using chatbots in language learning would increase academic productivity | 3.2 | 1.013 | 3.2 | 0.963 |
| 8. (PU) Using chatbots would make language learning easier | 3.4 | 1.050 | 3.7 | 0.830 |
| 9. (PU) Using chatbots in language learning allows the learners to study outside of the classroom | 3.9 | 0.918 | 4.2 | 0.609 |
| 10. (PU) Using chatbots in language learning is useful for context-based interactions as in real life | 3.4 | 1.092 | 3.7 | 1.054 |
| 11. (PU) Chatbots enable students to learn more quickly in language learning | 3.4 | 1.015 | 3.5 | 0.827 |
| 12. (PU) Chatbots make it easier to innovate in language learning | 3.6 | 0.934 | 3.7 | 0.716 |
| 13. (PU) The advantages of chatbots in language learning outweigh the disadvantages | 3.3 | 0.955 | 3.3 | 0.978 |
| 14. (US) I believe that using chatbots will increase the quality of language learning | 3.3 | 0.969 | 3.6 | 0.781 |
| 15. (PBC) I am completely satisfied in using chatbots for language learning | 3.0 | 1.038 | 2.7 | 1.015 |
| 16. (PBC) I am very confident in using chatbots in language learning | 3.0 | 0.991 | 3.2 | 0.933 |
| 17. (AT) Using chatbots in language learning is a good idea | 3.5 | 0.926 | 3.8 | 0.703 |
| 18. (AT) I am positive towards using chatbots in language learning | 3.4 | 1.010 | 3.6 | 0.827 |
| 19. (AT) Using chatbots in language learning is fun | 3.6 | 1.086 | 3.7 | 0.960 |
| 20. (BI) I intend to use chatbots in language learning frequently | 2.7 | 1.075 | 2.6 | 1.012 |
| 21. (BI) I intend to learn more about using chatbots in language learning | 3.0 | 1.107 | 3.2 | 1.075 |
| 22. (SE) I feel confident in using chatbots in language learning | 3.2 | 1.022 | 3.2 | 0.994 |
| 23. (SE) I have the necessary skills for using chatbots in language learning | 3.7 | 0.999 | 4.0 | 0.617 |
| 24. (PI) I like to experiment with new technologies in language learning | 3.8 | 0.995 | 4.0 | 1.071 |
| 25. (PI) Among my peers, I am usually the first to explore new technologies | 2.9 | 1.133 | 2.6 | 1.240 |
| Total | 3.4 | 0.996 | 3.5 | 0.878 |

Similarly, the Mann–Whitney U test was used to correlate the general perceptions (TAM2) concerning chatbot integration in language learning with gender and educational setting. No statistically significant differences ($p \leq 0.05$) were observed, as shown in Table 8.

**Table 8.** Effect of educational setting and gender on the TAM2 results.

| Gender | Mann–Whitney U | Z | Sig. (2-Tailed) | Educational Setting | Mann–Whitney U | Z | Sig. (2-Tailed) |
|---|---|---|---|---|---|---|---|
| TAM2 | 13,848,500 | −1.699 | 0.089 | TAM2 | 6,822,000 | −0.573 | 0.567 |

Qualitative data were collected through a template analysis (TA) available in Moodle (ANNEX C in Supplementary Materials). The thematic analysis was performed through the QDA Miner software following a deductive method based on predetermined codes. These were the same set of codes used in the ad hoc model designed (CHISM) and included in the post-survey as ordinal data (Likert scale). However, they were now arranged in two categories: benefits and limitations of each conversational partner. Therefore, the teacher candidates needed to explain these two categories in their own words, illustrate their insights with real examples and screenshots of their chatbot interaction, and submit the TA to the instructors during the last day of class. Table 9 shows the frequency of the most relevant codes for each conversational partner using QDA Miner software. After considering the participants' responses, it became necessary to add two new codes to our analysis: intentional meaning (pragmatics) and data privacy (design).

**Table 9.** Benefits and limitations of each conversational agent (code frequency).

| Chatbot | Category | Code (CHISM Item/s) | Freq. |
|---|---|---|---|
| Replika | Benefits | 1. natural conversation (semantic coherence & response interval) | 75.5% |
| | | 2. avatar can be customized (design) | 75.0% |
| | | 3. use of inclusive language and design, politeness (design & lexical richness) | 74.0% |
| | | 4. rich vocabulary, colloquialisms (lexical richness) | 73.0% |
| | | 5. storage capacity, memory, keeps a diary (design & interface) | 72.5% |
| | Limitations | 6. privacy issues, asks for personal information (data privacy) | 54.0% |
| | | 7. chatbot did not correct mistakes (error correction) | 42.5% |
| | | 8. chatbot made some grammar mistakes (grammatical accuracy) | 21.0% |
| Kuki | Benefits | 1. wide vocabulary options, colloquial expressions (lexical richness) | 75.0% |
| | | 2. coherent responses, no off topics (semantic coherence) | 64.5% |
| | | 3. a good number of social media options (multimedia content) | 64.0% |
| | | 4. rich variety of gifs, memes and emojis (non-verbal language) | 63.0% |
| | | 5. immediate response (response interval) | 52.5% |
| | Limitations | 6. bad sense of humor, very sarcastic, offensive (intentional meaning) | 54.0% |
| | | 7. insistence on upgradable pay options (design and data privacy) | 52.0% |
| | | 8. some off topics and misunderstandings (semantic coherence) | 21.0% |
| Wysa | Benefits | 1. innovative and helpful tips about mental issues (design) | 66.5% |
| | | 2. friendly avatar, penguin (design) | 45.0% |
| | | 3. user can exercise and play games or learn new things (multimedia) | 38.5% |
| | | 4. no error correction but does not hinder communication (error correction) | 35.0% |
| | | 5. positive use of pics and emojis (non-verbal language) | 34.5% |
| | Limitations | 6. some responses based on preset options, predictable (natural interaction) | 59.5% |
| | | 7. poor voice technology, canned (R& S technologies) | 45.0% |

Quantitative results seemed to be confirmed by the qualitative findings. Table 10 shows some comments selected from the template analysis (TA). Customizing options, semantic coherence and lexical richness became key factors for the teacher candidates' satisfaction. The three conversational partners selected had not been specifically programmed for language learning, so error detection and correction was not necessarily one of their functionalities. However, chatbot–human communication was not interrupted despite the presence of intentional mistakes or the use of abbreviations (P23 in Table 10). Still, some participants believed this lack of error correction could pose a problem for learners with a lower English level (P102), as already pointed out in previous works [47].

**Table 10.** Teacher candidates' insights about the conversational partners (TA selection).

| Subject | Chatbot | Comment (QA) |
|---|---|---|
| P45 (f) | Replika (pro) | Customizable avatar (you can decide their gender/sex, with the option of non-binary as well, very progressive, no apparent gender bias once they are talking to you: "I'm pro-choice" and "I support the right of a woman to make her own mind up", my male Replika answered when I asked him about his stance on feminism. |
| P102 (m) | Replika (con) | Interaction not interrupted by minor mistakes concerning capitalization, apostrophes, abbreviations ("sth" = something), but not corrected, so this could be counterproductive for some learners (mistakes go unnoticed by people using the chatbot to practice/learn English). |
| P23 (f) | Kuki (pro) | Mistakes are not corrected, but interaction is not hindered: to my "who's your favorite singer?" she sent me a picture of a French singer (Matthieu Tota) and told me briefly about him and shared some links on Spotify and YouTube. I think the use of multimedia content can be very positive for language learners. |
| P177 (f) | Kuki (con) | Similar responses containing same key words trigger the same responses on Kuki's part, so interaction feels sometimes repetitive and unexciting. For example, when I wrote "no worries" and "don't worry" (meaning "it's ok") at different times, she answered on both occasions: "People worry about things all the time, but I don't feel emotions like you humans do." This is loss of naturality, inability to understand language used figuratively (to my "I see" meaning "I understand", she answered "Where do you see it?") |
| P28 (f) | Kuki (con) | Kuki's sense of humor was weird; I found her responses sometimes offensive. For example, I made a grammatical mistake on purpose when she told me 'I like dance music' and she showed me her playlist, my answer was 'wow you HAD a playlist' and she responded 'I'm glad you're impressed. Yes, I often listen to music at Spotify, especially dance music. I think you mean 'You have' rather than 'You has'. Were you away from school that day?' And on another occasion, I tried to repeat her answers, she noticed it and said 'are you copying my answers for a reason?', to which I repeated the same and then she replied 'You can't be too intelligent if you have to repeat what a robot says'. |
| P189 (f) | Wysa (pro) | Wysa shows empathy and emotions, which makes conversations enjoyable and calm. Plus, as it stores information of what you tell, you can pick up where you left it, and that contributes to the feeling of reality in interaction. |
| P74 (m) | Wysa (con) | The voice sounds quite unnatural, like robotic or canned (R&S technologies), so I opted for the text-based option which seemed more natural to me. |

Similarly, the use of non-verbal language in the conversation, such as emojis and memes, and multimedia content, for example videos, enhanced the participants' engagement (P23), which confirms the multimodal nature of modern computer-mediated communication (instant messaging, social networking). This aspect may be particularly relevant for the younger generations and it has clear implications for future chatbot designers since potential users will expect a multimodal communication.

Concerning gender, female participants seemed to be more attentive to and appreciative of the use of an inclusive design and language. The thematic analysis results revealed that they asked more questions about gender and social minority issues to the conversational partners than their male counterparts (P45). This may confirm very recent research about chatbot design and gender stereotyping indicating that most voice-based conversational agents (Alexa, Cortana, etc) are designed to be female [49,50]. In general, the female participants were very assertive and concerned about the risk of replicating such gender stereotypes from the real to the digital world.

Data privacy became a major issue for a certain number of participants as some chatbots, particularly Replika and Kuki, repeatedly asked them to grant permission to access their social networks, use of cams for video calls, etc. Most participants showed concern about personal data storage and manipulation and therefore refused to grant such permission. They also showed this could be a major limitation for the use of chatbots among young language learners. In this case, the implication for chatbot developers is that

data protection and safety need to be prioritized in chatbot design, particularly if they are addressed to young learners or used in education.

Humor perception became a matter of controversy, particularly in the interaction with Kuki. Although the use of humor and intentional meaning (pragmatics) is still a challenge [51], the three conversational partners made use of it in different ways. In this respect, participants' overall satisfaction with Replika and Wysa was positive. However, a certain number of teacher candidates were surprised and even disturbed about kuki's use of humor (P28), which they found at times inappropriate or derogatory, and believed this could become a disadvantage for some language learners. In fact, decreased language anxiety has been found to be one of the main affordances about the use of such conversational AI among learners on the grounds of their non-judgmental character [17,19]. However, the use of humor is an advanced human skill and Kuki was featuring the conversation of an 18-year-old avatar as the teacher candidates were explained.

## 6. Conclusions and Implications

The use of AI in language learning is on the rise and chatbots are a good example. In this mixed-method research about the knowledge, level of satisfaction and perceptions concerning chatbot integration among future educators, three main conclusions and implications can be drawn. First, they have a low knowledge about modern chatbots which is mostly limited to the use of some Intelligent Personal Assistants (IPAs) and little experience in some areas like e-commerce. Consequently, there is a gap between language teacher candidates' preparation and recent advances in the application of AI to language learning which needs to be addressed with better training in the EFL curriculum. Future educators need to be well informed about the different types of chatbots as well as their benefits and limitations in language learning.

Secondly, both linguistic and technological features need to be considered for the successful integration of conversational agents as evidenced by the results of the ad hoc model (CHISM). Lexical richness and semantic coherence were important factors but other less often cited aspects in modern research played a major role, i.e., chatbot adaptivity to changing conditions and personalization. This partly confirms previous results that automated agents need to adapt their language to the proficiency level of their interlocutors [21] but future chatbots need also to be highly adaptive in terms of design and multimedia features. The quantitative analysis demonstrated that gender and educational setting had no effect on participants' satisfaction with the linguistic level of the conversational partners but the qualitative findings unveiled some gender-based differences regarding customizing options and topics of interaction. Female participants were more attentive to the use of inclusive design and language, and more assertive about gender stereotyping. Data storage and privacy also became a key issue for teacher candidates, with the implication that privacy needs to be carefully considered if chatbots are to be used among young language learners.

Thirdly, the TAM2 results demonstrated positive outcomes in perceived usefulness, easiness and attitudes toward chatbot integration but a moderate interest in using them in the near future. Human to human communication is prioritized among future educators although they are willing to learn more about this breakthrough technology. Multimedia content, gamification and use of non-verbal language could become key factors for learners' satisfaction as indicated by the teacher candidates. EFL students may not be willing just to chat with a bot, they will also expect to interact with a comprehensive partner that can converse, entertain and provide any supplemental information whenever it is required. Customized interaction, multimodal communication and integration with social media and other emerging technologies will also become essential features for chatbot success.

Finally, two main areas of improvement for its adoption in language learning are pragmatics and R&S technologies. There has been some recent progress but the shortcomings are clearly perceived as limitations among teacher candidates. On the one hand, contextual meaning and use of humor are essential aspects for user satisfaction. On the other hand, speech technologies (R&S) are the basis for successful human-like communication between



a bot and a person. In this experiment, teacher candidates preferred text-based over voice-enabled interaction with the chatbots when they sounded too robotic and unnatural, which is consistent with previous results [22].

This research has some limitations, a longitudinal study may be useful to better determine participants' perceptions and level of satisfaction after a longer period of interaction. Moreover, applying the ad hoc model used in this research (CHISM) to other educational contexts and with different conversational AI is necessary to compare the results with those presented in this study. Future directions include investigating teacher candidates' perceptions toward chatbot integration in EFL in terms of language and design adaptivity, ethics and privacy, as well as gender-related attitudes, following the gender-related differences unveiled regarding participants' satisfaction with chatbot design and topics of interaction.

**Supplementary Materials:** The following supporting information can be downloaded at: https://www.mdpi.com/article/10.3390/app12178427/s1. ANNEX A: Pre survey. ANNEX B: Post survey. ANNEX C: Template analysis—TA.

**Author Contributions:** Conceptualization, J.B.-M. and J.R.C.-F.; methodology, J.B.-M.; software, J.B.-M. and J.R.C.-F.; validation, J.B.-M. and J.R.C.-F.; formal analysis, J.B.-M.; investigation, J.B.-M.; resources, J.B.-M.; data curation, J.R.C.-F.; writing—original draft preparation, J.B.-M.; writing—review and editing, J.R.C.-F.; visualization, J.B.-M.; supervision, J.R.C.-F.; project administration, J.B.-M.; funding acquisition, J.B.-M. All authors have read and agreed to the published version of the manuscript.

**Funding:** This study is part of a larger research project, [The application of AI and chatbots to language learning], financed by the Instituto de Ciencias de la Educacion at the Univesity of Alicante (Reference number: 5498).

**Institutional Review Board Statement:** The study was conducted in accordance with the Declaration of Helsinki, and following the regulations in both institutions for studies involving humans, the University of Alicante (Spain) https://bit.ly/3yoLb05, and the Silesian University of Technology (Poland) https://bit.ly/3NTz6Wr.

**Informed Consent Statement:** Written informed consent was obtained from all participants involved in the study and all personal data which was not relevant to the study were anonymized.

**Data Availability Statement:** The data that support the findings of this study are available from the corresponding author. The data are not publicly available due to some personal data included in the human-chatbot interactions.

**Conflicts of Interest:** The authors declare no conflict of interest.

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
