# Peer review of "Using Chatbots as AI Conversational Partners in Language Learning"

_applsci, doi:10.3390/app12178427_

Round 1

Reviewer 1 Report

Review of the Manuscript

"Using chatbots as AI conversational partners in Language Learning."

The authors examine knowledge, level of satisfaction, and perceptions toward integrating conversational AI in language learning among future educators.

There are many studies about technical performance in general, but not so much about how people feel about them. The problem under investigation is current. Many big companies are invested/investing in chatbots and looking at real applications. I think their "Chatbot-Human interaction satisfaction model" can be applied to other areas of application of chatbots, and so this may be an important reference in the area.

The introduction, historical background,  results, etc. Are well written. The paper, in general, is well written and, in my opinion, engaging. The statistical analysis is sound.

The model proposed to measure the level of satisfaction of the participants is very interesting, and it seems to me that it can be adapted to other problems or used as part of a more generic model.

In my opinion, the manuscript should be accepted.

Author Response

Dear reviewer,

Thanks for the comments.

Best wishes,

The authors

Reviewer 2 Report

Minor typos (search by):

quizbot, cobot, digital assistant … => quizbot, digital assistant … // to remove “cobot” (it means “collaborative robot”)

3.1. Context and … => 4.1 Context and …
3.2. Materials … => 4.2 Materials ...

3.3. Method … => 4.3 Method …

Author Response

Dear reviewer,

Thanks for your comments. Please, find below our response:

1. Following your suggestion, “cobot” has been removed in line 64.
 quizbot, cobot, digital assistant … => quizbot, digital assistant … // to remove “cobot” (it means “collaborative robot”)

2. Following your suggestion, the numbering has been fixed.
3.1. Context and … => 4.1 Context and …
3.2. Materials … => 4.2 Materials ...
3.3. Method … => 4.3 Method …

Thanks once again for your time and feedbac.

The authors

Reviewer 3 Report

The paper presents a learning module about AI and language learning. The paper is well written with detailed background, discussion on the designed module, and results and discussion. Some minor points need to be addressed.

In the abstract, the authors state that “The findings also unveiled some gender-related differences regarding participants’ satisfaction with chatbot design and topics of interaction”. The results and discussion section should cover this point in more detail.

 Figure 1 needs to be renamed to Table 1.

 The column heading “gender”, in table 5 is confusing. Needs elaboration either in text or table caption.

Author Response

Dear reviewer,

Thanks for reading our manuscript and your positive feedback. Please, find below our responses:

  1. Regarding the observed gender-related differences in more detail, we  wish to indicate that, although interesting, this issue is out of bounds in relation to the scope of the paper and we are limited to paper length. We are actually working on another research work about gender issued and chatbot usage from a pragmatic perspective. However, following your suggestion, reference to it has been made in the discussion section in hopes to pave the way for future research in the area: “Future directions include investigating teacher candidates’ perceptions toward chatbot integration in EFL in terms of language and design adaptivity, ethics and privacy, as well as gender-related attitudes, following the gender-related differences unveiled regarding participants’ satisfaction with chatbot design and topics of interaction. “
    2.  Following your suggestion, Figure 1 has been renamed to Table 1.
    3. Following your suggestion, the heading “gender” has been changed to “participants’ gender” and figures have been provided in the text: “Generally, female participants (n=142) seemed to be more perceptive than male participants (n=34) about the customizing options of some chatbots as later explained in the qualitative results.”

We want to thank you once again for your time and valuable comments.

The authors
